# Killing of *Plasmodium* Sporozoites by Basic Amphipathic α-Helical Fusion Peptides

**DOI:** 10.3390/microorganisms12030480

**Published:** 2024-02-27

**Authors:** Manuela C. Aguirre-Botero, Eduardo Aliprandini, Anisha Gladston, Olga Pacios, Rafael Miyazawa Martins, Jean-Luc Poyet, Rogerio Amino

**Affiliations:** 1Institut Pasteur, Université Paris Cité, Malaria Infection and Immunity, BioSPC, 75015 Paris, France; 2Department of Life Sciences, Imperial College London, London SW7 2AZ, UK; 3INSERM UMRS976, Institut de Recherche Saint-Louis, Hôpital Saint-Louis, 75010 Paris, France; 4Université Paris Cité, 75012 Paris, France

**Keywords:** *Antimicrobial peptides*, penetratin, AAC-11, malaria, sporozoites, cytotoxicity

## Abstract

Membranolytic molecules constitute the first line of innate immune defense against pathogenic microorganisms. *Plasmodium* sporozoites are potentially exposed to these cytotoxic molecules in the hemolymph and salivary glands of mosquitoes, as well as in the skin, blood, and liver of the mammalian host. Here, we show that sporozoites are resistant to bacteriolytic concentration of cecropin B, a cationic amphipathic antimicrobial insect peptide. Intriguingly, anti-tumoral cell-penetrating peptides derived from the anti-apoptotic protein AAC11 killed *P. berghei* and *P. falciparum* sporozoites. Using dynamic imaging, we demonstrated that the most cytotoxic peptide, called RT39, did not significantly inhibit the sporozoite motility until the occurrence of a fast permeabilization of the parasite membrane by the peptide. Concomitantly, the cytosolic fluorescent protein constitutively expressed by sporozoites leaked from the treated parasite body while To-Pro 3 and FITC-labeled RT39 internalized, respectively, binding to the nucleic acids and membranes of sporozoites. This led to an increase in the parasite granularity as assessed by flow cytometry. Most permeabilization events started at the parasite’s posterior end, resulting in the appearance of a fluorescent dot in the anterior part of sporozoites. Understanding and exploiting the susceptibility of sporozoites and other plasmodial stages to membranolytic molecules might foster strategies to eliminate the parasite and block its transmission.

## 1. Introduction

*Plasmodium* sporozoites are formed inside extracellular oocysts growing on the abluminal side of the mosquito midgut. After release in the hemolymph, sporozoites invade and mature inside salivary glands until transmission to a vertebrate host [1]. During an infectious mosquito bite, this infectious and motile plasmodial stage is deposited in the host skin [2,3], where it searches for and invades a blood vessel at privileged sites associated with pericytes [4]. Once in the blood circulation, sporozoites specifically arrest in the liver [5] to cross the sinusoidal barrier [6,7] and infect hepatocytes. Inside these cells, one sporozoite transforms into thousands of erythrocyte-invasive stages, which are released back into the circulation to start blood infection, leading to malaria [8,9]. In this long journey from oocysts to hepatocytes, extracellular hemolymph and salivary gland sporozoites spend many days exposed to the mosquito immune system, while cutaneous, blood-circulating, and hepatic sporozoites are exposed for a few hours to the mammalian host humoral immune effectors.

Antimicrobial cationic peptides constitute an important innate immune defense against microorganisms. The cationic and amphipathic nature of these peptides is thought to be associated with the preferential binding and permeabilization of the negatively charged membranes of target cells [10,11].

Many natural antimicrobial peptides (AMPs), such as magainin, cecropin, and defensins, can kill oocysts from different plasmodial species when injected [12,13] or over-expressed in the insect hemolymph [14]. However, avian-infecting *P. gallinaceum* sporozoites are resistant to incubation with 250 µM of the AMPs metalnikowin, thanatin, drosocin, and metchnikowin. At this concentration, only dragonfly and fleshfly defensins were able to permeabilize sporozoites, killing 50% of parasites at 5 µM [13]. Yet, despite rendering mosquitoes susceptible to Gram-positive bacterial challenge, the knockdown of *Anopheles gambiae* defensin did not affect *P. berghei* infection, indicating that this AMP is not essential in controlling parasite density inside the mosquitoes reared and infected in laboratory conditions [15].

We have recently shown that the major sporozoite surface protein, the circumsporozoite protein (CSP), could shield the parasite membrane from the cytotoxic activity of a pore-forming protein secreted by sporozoites [16]. Here, we investigated the sporozoite’s susceptibility to different membrane-active molecules, focusing on the cytotoxic permeabilizing activity of basic amphipathic helical peptides.

## 2. Materials and Methods

### 2.1. Parasites, Mice, and Mosquitoes

In this study, *P. yoelii* 17XNL (Py) [17] and *P. berghei* ANKA (Pb) expressing GFP [18] or mCherry (Pb mCherry) [19] under the control of the hsp70 promoter and *P. falciparum* NF54 (Pf) [20] were used. *Anopheles stephensi* (SDA 500 strain) and *Anopheles gambiae* mosquitoes were reared at the Center for Production and Infection of Anopheles (CEPIA) at the Institut Pasteur in Paris. Mosquitoes were infected by blood-feeding on parasitized RjOrl:Swiss mice and maintained at 21 °C in a humid chamber with sucrose. Sporozoite suspension was obtained after manual dissection of infected salivary glands, homogenization in PBS, and filtration using a cell-strainer (35 µm mesh, BD Falcon). Sporozoite concentration was determined using Kova slides (Kova International). Mice were purchased from Janvier Labs and kept in the animal facility of the CEPIA at the Institut Pasteur in Paris, accredited by the French Ministry of Agriculture for performing experiments on rodents. Parasites were obtained from the salivary glands of infected mosquitoes isolated 19–26 days after infection.

### 2.2. Peptides 

Peptides (purity > 95% HPLC) were synthesized by Proteogenix (Strasbourg, France). The peptides sequences are: RK16: RQIKIWFQNRRMKWKK, RT53: RQIKIWFQNRRMKWKKAKLNAEKLKDFKIRLQYFARGLQVYIRQLRLALQGKT, RL 37: RQIKIWFQNRRMKWKKKDFKIRLQYFARGLQVYIRQL, RT39: RQIKIWFQNRRMKWKKLQYFARGLQVYIRQLRLALQGGKT, RT39M: RQIKIWFQNRRMKWKKLQYFAAGLQVYIRQLRLALQGGKT, RT39-FITC: FITC-RQIKIWFQNRRMKWKKLQYFARGLQVYIRQLRLALQGGKT, and RT39M-FITC: FITC-RQIKIWFQNRRMKWKKLQYFAAGLQVYIRQLRLALQGGKT. Cecropin B from *Hyalophora cecropia* (purity ≥ 97% HPLC) was purchased from Sigma-Aldrich (Burlington, NJ, USA)

### 2.3. Cytotoxicity Assays

To assess the cytotoxic effect of the different peptides on sporozoites, we performed an in vitro assay as previously described [21]. Briefly, sporozoites were incubated at 37 °C for 45 min with different peptides in the presence of 10% fetal calf serum (FCS). Samples were next, incubated for 10 min with 5 µg/mL propidium iodide (PI, Invitrogen, WA, USA) on ice and either imaged in an inverted Axio Observer Z.1 microscope (Zeiss, Oberkochen, Germany) and analyzed using Fiji [22] or diluted 10 times with cold PBS before acquisition on a CytoFLEX S flow cytometer (Beckman Coulter, Brea, CA, USA). For the Pf cytotoxic assay, a final concentration of 0.5 µM cytochalasin D (Sigma-Aldrich) was used in some conditions. Viability was defined as the percentage of GFP+PI- sporozoites to the sum of GFP+PI-, GFP+PI+, and GFP-PI+ sporozoites. Data were analyzed using the CytExpert 2.0 software (Beckman Coulter). 

*Escherichia coli DH5α* (Ec) were grown at 37 °C under agitation in LB medium at an OD_620_ of approximately 0.35 (corresponding to 10^8^ CFU/mL). After centrifugation and washing with PBS, the equivalent of 10^7^ colony-forming units (CFUs)/mL were incubated at 37 °C for 10 min with or without Cecropin B (Sigma-Aldrich) in 10%FCS/PBS at concentrations ranging from 2 to 100 μM. The suspensions of bacteria were diluted 1000× in PBS and 50 µL were spread on LB-agar plates for CFU quantification. Viability was assessed by normalizing the number of CFU to the control group. A sample of *E. coli* heated at 70 °C for 10 min was used as a death control.

### 2.4. Dynamic Microscopy

To visualize the killing of the sporozoites by the peptide, Pb sporozoites expressing mCherry were incubated with 20 μM of RT39 and RT39-FITC. First, 5000 sporozoites were activated in 10% FCS/PBS on an 18-well slide (iBidi, Gräfelfing, Germany). After 5 min, the peptide was added with 1 μM of To-Pro 3 and the sporozoites were immediately recorded for 15–30 min using a spinning-disk confocal system (UltraView ERS, PerkinElmer, Shelton, CT, USA) controlled by Volocity (PerkinElmer), composed of 4 Diode Pumped Solid State Lasers, a Yokogawa Confocal Scanner Unit CSU22, a z-axis piezoelectric actuator, and a Hamamatsu Orca-Flash 4.0 camera mounted on a Axiovert 200 microscope (Zeiss). Time-lapses were analyzed using Fiji [22].

### 2.5. Gliding Motility Assay 

To assess gliding motility, 5000 sporozoites were resuspended in 10% FCS/PBS in a final volume of 10 µL. The resulting suspension was transferred to an 18-well slide (iBidi) and centrifuged at 400× *g* for 3 min at 4 °C. The slide was then allowed to equilibrate at 37 °C, 5% CO_2_ for 3 min in the incubation chamber (Incubation System S, Zeiss) of an inverted epifluorescence wide-field microscope (AxioObserver Z.1, Zeiss) equipped with a LED illumination system (Colibri2, Zeiss), a CCD camera (AxioCam MR, Zeiss), and controlled by the AxioVision software (version 4.8.2.0, Zeiss). After 3 min, a time-lapse movie at a rate of one image per second was recorded for 2 min to control for initial parasite motility. After the 2 min recording, 10 µL of PBS or 20 µM of RT39 were added on top, and the parasites were immediately imaged for 20 min. The average sporozoite velocity at different time points was determined using the MTrack2 plug-in from Fiji [22].

### 2.6. Statistical Analysis 

Statistical significance was determined with a one-way ANOVA with Holm–Šídák correction for multiple comparisons with help of GraphPad Prism 9. 

## 3. Results

### 3.1. Effect of Cationic Amphipathic Peptides on Sporozoites

To evaluate the killing of plasmodial sporozoites by membrane-active molecules, we tested the activity of cationic amphipathic α-helical peptides listed in Figure 1 on *P. berghei* (Pb), *P. yoelii* (Py), and *P. falciparum* (Pf) salivary gland sporozoites in vitro. Quantification of sporozoite death was measured by flow cytometry based on the uptake of propidium iodide (PI), a live cell impermeant nucleic acid fluorescent dye, and/or by the loss of parasite fluorescence following 45 min incubation at 37 °C in PBS 10% fetal calf serum (FCS) [16].

Cecropin B is a prototypical antimicrobial peptide (AMP, Figure 1) discovered in the hemolymph of the Cecropia moth [23]. This positively charged AMP is known to inhibit the development of *P. knowlesi*, *P. cynomolgi,* and *Pf oocysts* in vivo at concentrations that start to be toxic to mosquitoes [12]. Since our sporozoite killing assay contains 10% FCS, which can inhibit the antimicrobial activity of several AMPs [24], we first tested the antimicrobial activity of cecropin B in the presence of 10% FCS. As shown in Figure 2A, incubation of *E. coli* (Ec) with 2 µM of cecropin B for 10 min at 37 °C was enough to efficiently kill bacteria in this condition. Contrarily, cecropin B did not significantly kill Pb and Py sporozoites at 100 µM over 45 min of incubation.

We next tested the killing activity of a series of cationic amphipathic peptides consisting of penetratin (Figure 1, RK16), a cell-penetrating peptide (CPP) derived from the *Drosophila melanogaster* homeotic protein antennapedia a-helix 3 [25], fused with peptides derived from the a-helix 18 of the antiapoptosis clone 11 (AAC11) also known as Apoptosis Inhibitor 5 protein [26] (Figure 1, RL37, RT39, and RT53). RT53 and RT39 act as decoys that can prevent interaction between AAC11 and its binding partners and are known to elicit the cell death of cancer cells [27,28] and of CD4+ T cells susceptible to HIV infection [29]. Both peptides adopt an extended helical structure that includes aromatic amino acids and a hydrophobic face (Figure 1B). At a concentration of 20 µM, penetratin alone (RK16) or the fused peptide RL37 did not significantly kill GFP-expressing Pb sporozoites (Figure 2B). Conversely, RT53, which contains the longest sequence of the AAC11 a-helix 18, killed around 50% of Pb sporozoites. Surprisingly, RT39, a shorter peptide lacking the 14 first amino acids of the N-terminal part of the AAC11 a-helix 18, killed almost all parasites during 45 min of incubation at 37 °C (Figure 2B). The same profile of parasite killing was observed using Pf sporozoites (Figure 2C). In both cases, RT39 was as lethal as or more than 10 µg/mL of cytotoxic anti-CSP antibodies. However, differently from cytotoxic antibodies, peptide cytotoxicity was not dependent on the parasite motility [16]. Treatment of sporozoites with cytochalasin D, an actin-polymerizing inhibitor, at a concentration that completely blocks sporozoite motility, did not revert parasite death (Figure 2C). Analysis of granularity by flow cytometry indicated that sporozoites killed by antibody or peptides displayed a distinct side scatter profile from live sporozoites (Figure 2D, FSC × SSC graph). However, the PI fluorescence intensity of sporozoites killed by RT39 was slightly inferior to those killed by the anti-CSP antibody, suggesting that the positively charged peptide could interfere with the binding of PI to the negatively charged parasite nucleic acid (Figure 2D, scatterplot). Increased granularity of dead parasites was associated with increased refringence around a circular central structure by microscopy (Figure 2E).

**Figure 1 microorganisms-12-00480-f001:**
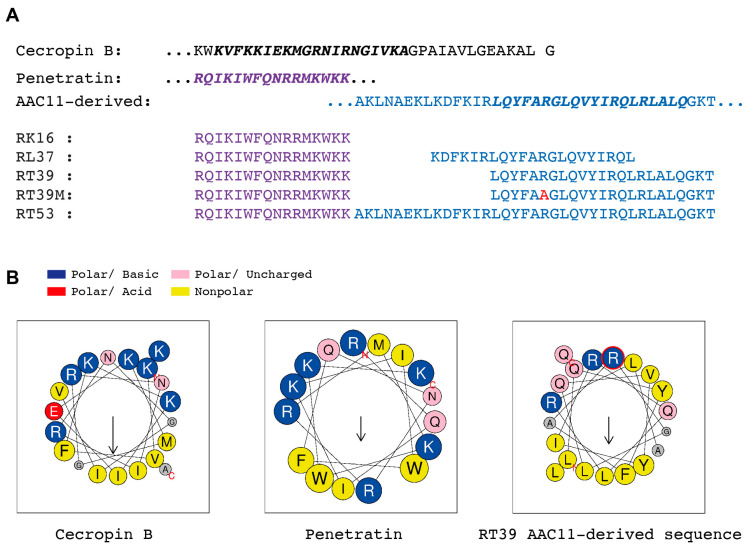
Structures of peptides used in this work. (**A**) Primary structure of *Hyalophora cecropia* cecropin B (Uniprot: P01508, CECB_27-61_), *Drosophila melanogaster* homeotic protein antennapedia a-helix 3, penetratin (Purple, Uniprot: P02833, ANTP_R339-K354_), and human AAC11 a-helix 18 (Blue, Apoptosis Inhibitor 5, Uniprot: Q9BZZ5, API5_HUMAN_A363-T399_). Red—Arginine to Alanine mutation in the AAC11-derived peptide sequence. (**B**) Projection of cationic amphipathic α-helical structures marked in bold italics in the sequences shown in A using Heliquest [30]. Notice the clear bipartition between polar and nonpolar amino acids in Cecropin B and AAC11-derived peptide wheels and the alternated distribution in penetratin. Arrows show the direction of the hydrophobic moment.

**Figure 2 microorganisms-12-00480-f002:**
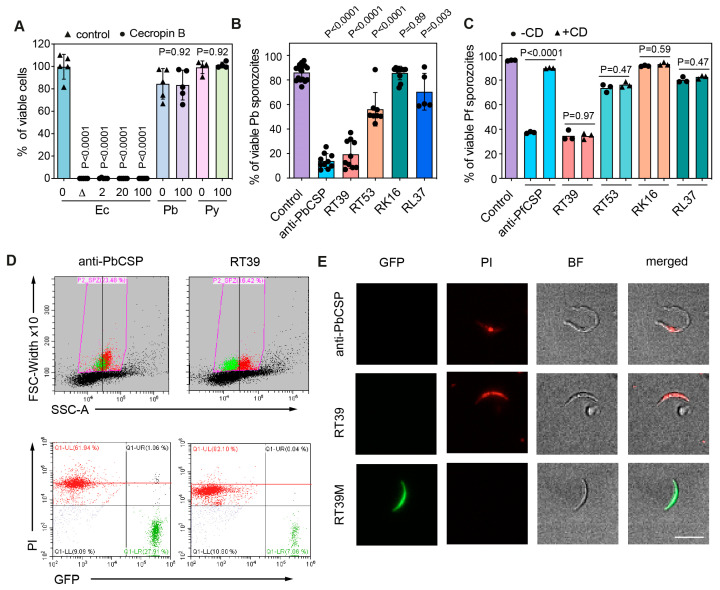
RT39 kills *Plasmodium* sporozoites. (**A**) Viability of *E. coli* (Ec), *P. berghei* (Pb), and *P. yoelii* (Py) sporozoites in the absence (0) or presence of 2, 20, or 100 μM cecropin B in 10% FCS/PBS. The graph shows the mean ± SD of pooled data from 3–4 independent experiments. D—heat-inactivated bacteria. Adjusted *p*-values show the comparison of Cecropin B-treated group with control groups and were obtained using ANOVA with Holm–Šídák correction. (**B**) Viability of Pb sporozoites after 45 min of incubation at 37 °C with 20 μM of different AAC11-derived peptides in 10% FCS/PBS. A measure of 10 μg/mL of a cytotoxic anti-PbCSP antibody (3D11) was used as control. The graph shows the mean ± SD of pooled data from 3–9 independent experiments. Comparison with the control mean using ANOVA with Holm–Šídák correction. (**C**) Viability of *P. falciparum* (Pf) sporozoites after 45 min of incubation at 37 °C with 20 μM of different AAC11-derived peptides in PBS 10%FCS, in the presence of cytochalasin D (CD), an actin-polymerizing inhibitor (+CD, gray squares), or absence (-CD—black circles). A measure of 100 μg/mL of a cytotoxic anti-PfCSP antibody (2A10) was used as control. Comparison of -CD and +CD group means using ANOVA with Holm–Šídák correction. (**D**) Sporozoite viability was measured by flow cytometry. Representative plots depict the gating strategy used to measure sporozoite viability and the length/granularity of Pb sporozoites after incubation with vehicle (control), anti-PbCSP (3D11), or RT39. Viability was defined as the percentage of GFP-positive sporozoites (green) that did not uptake propidium iodide (PI, red). (**E**) Representative images of the parasites after incubation with 20 μM of RT39, RT39M, and 10 μg/mL anti-PbCSP (3D11). BF—bright field. Scale bar = 10 μm.

### 3.2. Characterization of RT39 Mode of Action 

RT39 induced a fast killing of Pb sporozoites in comparison with anti-CSP antibody as measured by flow cytometry. While 20 µM of the cytotoxic peptide killed most parasites before 5 min of incubation, 10 µg/mL of cytotoxic antibody needed more than 20 min to eliminate the majority of sporozoites (Figure 3A). To better understand the way RT39 is acting on Pb sporozoites, we dynamically imaged the cytotoxic effect of the peptide on fluorescent sporozoites in the presence of To-Pro 3, a live-cell impermeant far-red fluorescent nucleic acid dye. Given its fast-killing kinetics, RT39 was added to a microscopy chamber containing PBS 10% FCS and sporozoites, which were immediately recorded for 20 min at 37 °C. In the presence of FCS, most sporozoites move in a circular way in a substrate-dependent manner [31]. Sporozoite death was assessed by the incorporation of To-Pro 3 and/or by the loss of cytoplasmic fluorescence. Following the addition of RT39 to sporozoites, a lag phase of around 5 min was observed, probably corresponding to the diffusion and peptide binding to sporozoites, after which parasites started to die in quick succession (Figure 3B). Two-minute analysis of sporozoite speed at the end of the lag phase or at the beginning of the death plateau when the number of parasites in the field did not substantially change (Figure 2B, 120 s interval between the dotted lines) showed that RT39 did not significantly impact sporozoite motility (Figure 2C). However, a sudden stop in motility (cyan line, ~450 s) simultaneous to increasing permeability to To-Pro 3 (gray line) and mCherry leakage (red line) was observed in the dying sporozoite population (exemplified in Figure 2D, ~450 s). The concomitance of To-Pro 3 entry and mCherry leakage as well as the parasite immobilization strongly suggested that a single event of membrane permeabilization took place leading to the three simultaneous phenomena.

The killing activity of RT39 was strongly dependent on the arginine (R6) from the AAC11-peptide sequence (Figure 1A,B, RT39 AAC11-derived sequence, red circle, R_22_ in the RT39 sequence). Substitution of this residue by an alanine (Figure 1A, alanine in red, RT39M), which is known to prevent interactions between RT39 and its binding partners [32], strongly decreased Pb sporozoite death (Figure 4A). However, this lack of cytotoxicity does not seem to be related to the inhibition of peptide binding to the parasite, since both FITC-labeled peptides displayed equivalent binding intensity to live Pb sporozoites (mCherry+/FITC^low^), as well as strong staining of dead parasites (mCherry-/FITC^high^, Figure 4B). We next dynamically imaged mCherry-expressing sporozoites in the presence of FITC-labeled RT39 and To-Pro 3 using high-speed spinning-disk microscopy. RT39-FITC also strongly labeled sporozoites simultaneously to increase parasite membrane permeability to To-Pro 3 and mCherry (Figure 4C,D). Differently from the cytometry data, where live sporozoites were weakly stained by RT39-FITC (Figure 4B, Cherry+/FITC^low^), live parasites observed by microscopy had the same level of background fluorescence, presumably due to the presence of 20 µM of fluorescent peptide in the solution (Figure 4C,D, Ratio MFI SPZ/Bk ~1). In a few cases, accumulation of RT39-FITC on the sporozoite membrane preceded permeabilization to To-Pro 3 (Figure 4E, posterior, 121 s). To-Pro 3 then diffused inside the sporozoite body, labeling the parasite nucleic acids, while RT39-FITC mainly stained the parasite membrane. Permeability to To-Pro 3 and RT39-FITC frequently occurred at the posterior part of the sporozoite body (Figure 4C III and Figure 4E, posterior, arrowhead), but not exclusively. This posterior permeabilization often led to a particular sporozoite death phenotype reminiscent of the dotty death induced by cytotoxic antibodies [16], but with the formation of the fluorescent dot at the anterior pole of parasites (Figure 4C III, asterisk and Figure 4E, posterior, 122 s). Spatial-temporal analysis of To-Pro 3 mean fluorescence intensity also showed permeabilization occurring at the central and anterior part of the sporozoite body (Figure 4E, central and anterior). Like the analysis by flow cytometry, dead sporozoites displayed much higher levels of RT39-FITC fluorescence than live parasites by microscopy. The fluorescent cytotoxic peptide accumulated in the blebs of dead parasite membrane, frequently close to the place of initial permeabilization (Figure 4E, last panel of the time-lapse).

## 4. Discussion

Metazoans rely on basic amphipathic α-helical peptides, known to preferentially bind to and permeabilize negatively charged membranes to eliminate microbes while preserving the integrity of their healthy cells [10]. Based on studies using model membranes, two main categories of mechanisms leading to the membrane permeabilization have been proposed: membrane curvature modulation and phase separation. Depending on the peptide-to-lipid ratio and lipid composition, membrane-active peptides can form toroidal or barrel–stave pores, permeabilize cells by carpeting the membrane and at high concentration, and act as detergents [10,11]. Here, we show that Pb and Py salivary gland sporozoites are resistant to bacteriolytic concentration of the cationic amphipathic cecropin B despite the susceptibility of plasmodial oocysts to high concentrations of this helical AMP [12]. Assuming oocysts and sporozoites have similar lipid membrane composition, the sporozoite resistance to cecropin could rely on the peptide hydrolysis by intramembrane parasite serine proteases [33]. For example, basic amphipathic α-helical peptides derived from trialysin [34], a voltage-dependent pore-forming protein from the saliva of the kissing bug *Triatoma infestans* are capable of killing the protozoans *Trypanosoma cruzi* [35] and *Acanthamoeba castellanii* [36], but these peptides can also be inactivated by amoebal serine proteases [36]. Interestingly, intramembrane and trypsin-like serine proteases preferentially cleave nonpolar residues in hydrophobic membrane domains [37] and substrates with basic residues at P1 [38], respectively, which are both distinctive features of cationic amphipathic AMPs. Alternatively, the protective shield offered by the major surface sporozoite protein, CSP, could specifically block peptide accessibility to the parasite membrane [16]. 

Conversely, RT53, which combines the sequences of two basic amphipathic helical peptides (Figure 1B), one capable of translocating into cells [25] and the other containing a heptad leucine repeat domain, which acts as a protein–protein interaction domain [32,39], was able to kill Pb and Pf sporozoites. In human cells, RT53 has an equivalent or higher membrane permeabilization activity than its shorter counterpart RT39 [29,32], plateauing after 60 min of incubation [28]. However, in plasmodial sporozoites, RT39 displayed a stronger killing activity than RT53 (Figure 2B,C), permeabilizing most parasites after 5 min of incubation (Figure 3A), indicating that RT39 kills sporozoites by a direct and fast permeabilization of the parasite membrane. 

Strong sporozoite staining with this peptide (MW = ~5 kDa) occurred together with the parasite permeabilization to To-Pro 3 (MW: 0.671 kDa) and mCherry (MW: 28 kDa). This singular temporal event which led to the transit of fluorophores of different sizes across the sporozoite membrane indicates the formation or the activation of a large pore on the parasite surface (Figure 3D and Figure 4D). RT39 possesses an amphipathic a-helix–loop–a-helix structure similar to that of numerous membrane-active, pore-forming peptides [40]. Therefore, it is possible that RT39 inserts and accumulates in sporozoite cell membranes upon binding to endogenous membrane partner(s), resulting in pore/carpet formation and membrane permeabilization. Interestingly, mutation of R_22_ in RT39 (peptide RT39M) decreased the pore-forming ability of the peptide. Previous analysis indicated that RT39 and RT39M possess a similar helical structure [32], ruling out mutagenesis-induced perturbation of the 3D structure that could impede RT39M membrane-active properties. A similar mutation in RT39 or AAC11 prevents their interaction with common protein partners [32], but since this mutation did not affect the binding of fluorescently labeled peptides to live or dead sporozoites it is unlikely that this residue plays a role in the interaction with partner(s) on the membrane and inside the parasite. Therefore, the decreased membrane permeabilizing activity observed with RT39M might be explained by a poor membrane-disrupting activity either linked to the peptide oligomerization or the pore channel/carpet formation. 

Sporozoite motility, which depends on Ca^2+^ mobilization [41], was also blocked by RT39 but only at the moment of membrane permeabilization to To-Pro 3 and mCherry (Figure 3D). The ionic imbalance occurring by the increased membrane permeability seems thus to be enough to block parasite motility and led sporozoites to death. 

Of It is of note that, despite their common basic amphipathic helical nature, cecropin B and RT39 differ in their basic and hydrophobic amino acid composition. While cecropin B is rich in lysine, asparagine, and isoleucine, the AAC11-derived peptide is rich in arginine, glutamine, and leucine (Figure 1). These differences could also be involved in the sporozoite susceptibility to these two types of peptides. Substitution of arginine by lysine in the basic amphipathic synthetic peptide, RR VW R WV RR VW R WV RR (e.g., it drastically changes its lytic activity towards bacteria and erythrocytes without significantly modifying its net charge, hydrophobicity, or hydrophobic moment) [42]. Glutamine, which is linked to the aggregation of glutamine-rich peptides [43], is also underrepresented in AMPs [11] but overrepresented in AAC11-derived peptides. The preferential posterior permeabilization of sporozoites by RT39, which seems to be independent of the CSP coat presence and motility, indicates the existence of points of susceptibility in the parasite body which could correspond to membrane domains containing specific partner molecules or lacking CSP or another inhibitory molecule. 

Sporozoites have a low salivary gland invasion efficacy, being rapidly degraded in the mosquito hemolymph [44]. However, this destruction is not correlated with phagocytosis by hemocytes [44] or defensin expression [15]. After salivary gland invasion, sporozoites accumulate in the gland lumen for several days [45], indicating an unimportant immune elimination inside this organ. It is known that the expression of AMPs in the mosquito midgut and hemolymph can reduce the number of sporozoites and moderately decrease the prevalence of sporozoite infection in the salivary glands of transgenic mosquitoes [46]. Since sporozoites seem to be relatively resistant to AMPs, RT39 ectopic expression in the mosquito hemolymph and salivary glands could thus constitute an efficient way to target sporozoites in transgenic mosquitoes to improve the control of parasite transmission. 

On the other hand, targeting sporozoites in the human skin, blood, and liver by the administration of membrane-active peptides seems to be more difficult due to the relatively high amount required per body weight, the temporal uncertainty of the parasite transmission, peptide biodistribution and degradation in the tissues of interest, and the shorter window of time to kill extracellular sporozoites. Studies using RT39 to target tumors in vivo [32,47] nonetheless support the study of the killing activity of AAC11-derived peptides against liver, blood, sexual, and other insect plasmodial stages. Many of these stages develop inside a parasitophorous vacuole; translocation through two lipid membranes will be required to directly reach the parasite membrane. The addition of CPP to AMPs is known to increase their antimicrobial activity, mainly against Gram-negative bacteria [48,49]. However, RT39 does not seem to translocate into sporozoites, the addition of CPP to AAC11-derived peptides or other AMPs could improve the killing efficacy of intracellular and extracellular plasmodial stages in vivo. A deeper understanding of the determinants underlying the efficient elimination of pathogens by membrane-active molecules is essential to better design strategies to control pathogen infection and transmission.

## Figures and Tables

**Figure 3 microorganisms-12-00480-f003:**
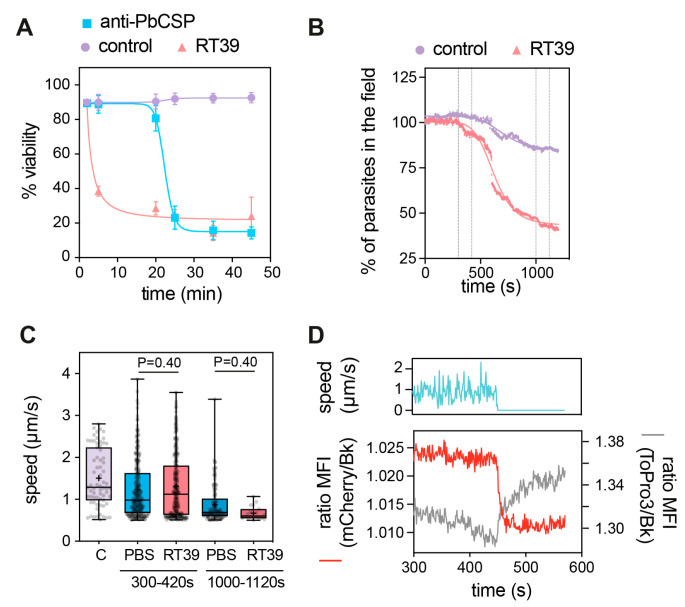
Fast killing of sporozoites by RT39. (**A**) Killing kinetics of RT39 in comparison to a cytotoxic anti-PbCSP antibody (3D11); Pb sporozoites were incubated for the indicated time with 20 μM peptide, 10 μg/mL antibody or PBS at 37 °C (n = 2 data are presented as mean ± SEM). Sporozoite viability at different concentrations of RT39 and RT39M (2, 20, 40, 80 μM) after incubation for 45 min at 37 °C. Viability was assessed by flow cytometry. (**B**–**D**) Analysis of the effect of 20 μM RT39 on parasite motility. Pb GFP sporozoites were recorded immediately after the addition of the peptide for 20 min at 37 °C with 5% CO_2_. (**B**) Percentage of parasites in the field over time during imaging. Death was characterized by the loss of the GFP signal. (**C**) Analysis of speed. For the control (C), the speed was quantified for 2 min before the addition of the peptide. After adding PBS or 20 µM RT39, the speed was analysed between 300 and 420 s and between 1000 and 1120 s. Each dot represents a parasite of 2–3 independent experiments. Box plots show the 25th to 75th percentiles, median, and mean (‘‘+’’), and the whiskers go to the minimum and maximum values. Adjusted P-values show the comparison of cecropin B-treated with control groups and were obtained using ANOVA with Holm–Sidak correction. (**D**) The graph shows side by side the speed, the GFP fluorescence, and of To-Pro 3 into the parasite before and at the moment of death. MFI—mean fluorescence intensity.

**Figure 4 microorganisms-12-00480-f004:**
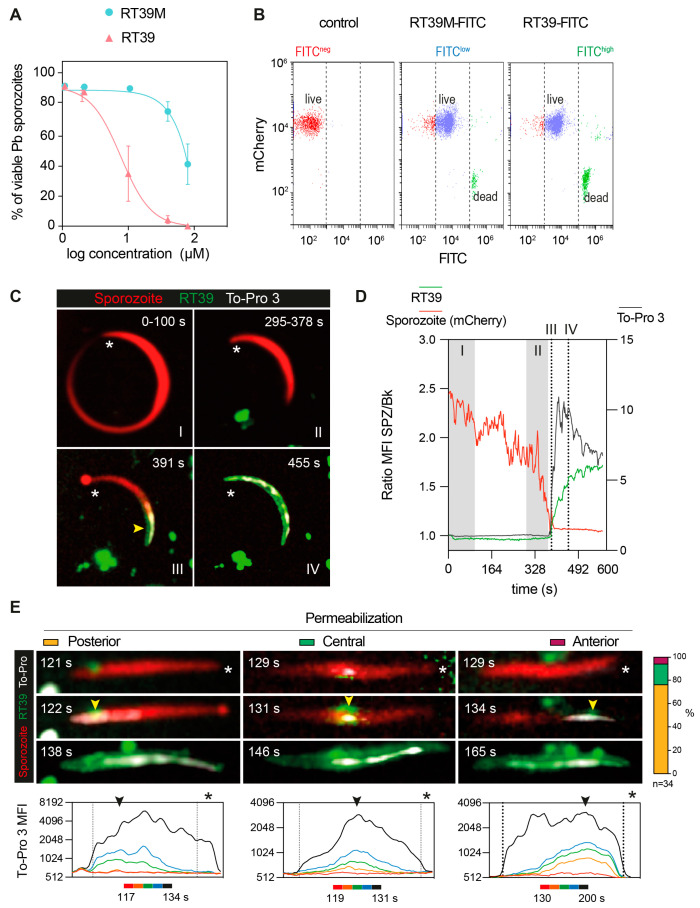
Dynamic imaging of sporozoite killing by RT39. (**A**) Sporozoite viability at different concentrations of RT39 and RT39M (2, 10, 40, 80 μM) after incubation for 45 min at 37 °C (n = 3 data are presented as mean ± SEM). (**B**) Representative plots of the binding assessment of 20 μM RT39-FITC and RT39M-FITC to Pb mCherry sporozoites by flow cytometry (n = 2–3). (**C**–**E**) Pb mCherry (in red) sporozoites in the presence of 20 μM RT39-FITC (in green) were recorded in the presence of 1 μM To-Pro 3 and 10% FCS/PBS. (**C**) Representative time-lapse from two independent experiments. The asterisk points to the anterior pole of the parasites and the arrow at the point of entry for the labeled peptide and To-Pro 3. (**D**) MFI—mean fluorescence intensity of RT39-FITC (green), To-Pro 3 (gray), and mCherry (red) inside the living (I), dying (II), and dead (III, IV) parasite. (**E**) Representative images of the different permeabilization phenotypes recorded. Arrowheads indicate the initial site of permeabilization and the asterisk points to the apical pole of the sporozoite. On the right, is the percentage of parasites that were categorized in each phenotype (n = 34 sporozoites from 2 independent experiments). Bottom graphs: time-lapses of the MFI profile of To-Pro 3 along the parasite body show the beginning of membrane permeabilization.

## Data Availability

Data are contained within the article.

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
