# Peer review of "Killing of Plasmodium Sporozoites by Basic Amphipathic α-Helical Fusion Peptides"

_microorganisms, 2024, doi:10.3390/microorganisms12030480_

Round 1

Reviewer 1 Report

Comments and Suggestions for Authors

In terms of originality and novelty, the study contributes new insights into the susceptibility of Plasmodium sporozoites to membranolytic molecules, particularly anti-tumoral cell-penetrating peptides derived from the anti-apoptotic protein AAC11. All the cited references appear to be relevant to the research, supporting the claims made in the article and providing context for the study. The use of dynamic imaging techniques to investigate the cytotoxic effects of peptides on Plasmodium sporozoites is well-suited to address the research questions posed by the study. The methods are adequately described, allowing for reproducibility and transparency in the research process. The conclusions drawn in the article are well-supported by the results obtained from the study. The authors effectively connect their findings to the broader implications for strategies aimed at eliminating the parasite and blocking its transmission. The manuscript is likely to be of interest to a wide range of readers, including researchers in the fields of immunology, parasitology, and infectious diseases. The figures are informative and presented in good resolution, with captions that are generally detailed. I believe the manuscript can be published in its current form or with minor revisions.

 Minor comments:

 Lines 93. In the sentence "Briefly, sporozoites were incubated at 37°C for 45 min with the different peptides in the presence of 10% FCS," consider specifying what "FCS" stands for to avoid any potential confusion.

Lines 112, 122. How were 5,000 sporozoites counted?

I think that the authors will also be interested in the results obtained in recent work, where the authors also used this peptide as a cell-penetrating peptide along with an antimicrobial peptide, and the resulting hybrid peptide showed less activity than other CPPs and with two CPPs (Int. J. Mol. Sci. 2023, 24(23), 16723; https://doi.org/10.3390/ijms242316723).

Author Response

We would like to thank the reviewers for their pertinent suggestions and remarks, as well as for their time to review our manuscript. Please see the attachment.

Reviewer 2 Report

Comments and Suggestions for Authors

The article "Killing of Plasmodium sporozoites by basic amphipathic α-helical fusion peptides" explores how certain peptides can target and eliminate Plasmodium sporozoites, which are critical stages in malaria transmission. The research focuses on the peptides' membranolytic activities, demonstrating that the peptides, especially RT39, can permeabilize sporozoite membranes, leading to their rapid killing. The research was carried out at a high level and is of undoubted interest. However, before publishing, I would recommend making the following minor changes:

I reccomend expand on the discussion on potential mechanisms of action of the peptides and how these relate to known pathways of Plasmodium infection and immunity.

I suggest conducting a more thorough comparison of the findings with existing literature, particularly other antimalarial compounds or vaccine strategies.

It would be beneficial to more explicitly address the limitations of your study, including potential challenges related to the in vivo degradation of peptides and the obstacles to efficient delivery at infection sites.

Author Response

(The authors gave the same response as above.)
